# An In Vivo Study of a Rat Fluid-Percussion-Induced Traumatic Brain Injury Model with [^11^C]PBR28 and [^18^F]flumazenil PET Imaging

**DOI:** 10.3390/ijms22020951

**Published:** 2021-01-19

**Authors:** Krishna Kanta Ghosh, Parasuraman Padmanabhan, Chang-Tong Yang, Zhimin Wang, Mathangi Palanivel, Kian Chye Ng, Jia Lu, Jan Carlstedt-Duke, Christer Halldin, Balázs Gulyás

**Affiliations:** 1Lee Kong Chian School of Medicine, Nanyang Technological University, 59 Nanyang Drive, Singapore 636921, Singapore; gkkanta@ntu.edu.sg (K.K.G.); yang.changtong@sgh.com.sg (C.-T.Y.); good_piggy@msn.com (Z.W.); PMAT0001@e.ntu.edu.sg (M.P.); christer.halldin@ki.se (C.H.); 2Cognitive Neuroimaging Centre, Nanyang Technological University, 59 Nanyang Drive, Singapore 636921, Singapore; 3Department of Nuclear Medicine and Molecular Imaging, Radiological Sciences Division, Singapore General Hospital, Outram Road, Singapore 169608, Singapore; 4Duke-NUS Medical School, 8 College Road, Singapore 169857, Singapore; 5DSO National Laboratories (Kent Ridge), 27 Medical Drive, Singapore 117510, Singapore; nkianchy@dso.org.sg (K.C.N.); ljia@dso.org.sg (J.L.); 6President’s Office, Nanyang Technological University, 50 Nanyang Avenue, Singapore 639798, Singapore; jan.carlstedt-duke@ntu.edu.sg; 7Department of Clinical Neuroscience, Karolinska Institute, S-171 76 Stockholm, Sweden

**Keywords:** traumatic brain injury (TBI), lateral fluid percussion (LFP), positron emission tomography (PET), neuroinflammation, GABA_A_-benzodiazepine receptor, PET radioligand, [^11^C]PBR28, [^18^F]flumazenil

## Abstract

Traumatic brain injury (TBI) modelled by lateral fluid percussion-induction (LFPI) in rats is a widely used experimental rodent model to explore and understand the underlying cellular and molecular alterations in the brain caused by TBI in humans. Current improvements in imaging with positron emission tomography (PET) have made it possible to map certain features of TBI-induced cellular and molecular changes equally in humans and animals. The PET imaging technique is an apt supplement to nanotheranostic-based treatment alternatives that are emerging to tackle TBI. The present study aims to investigate whether the two radioligands, [^11^C]PBR28 and [^18^F]flumazenil, are able to accurately quantify in vivo molecular-cellular changes in a rodent TBI-model for two different biochemical targets of the processes. In addition, it serves to observe any palpable variations associated with primary and secondary injury sites, and in the affected versus the contralateral hemispheres. As [^11^C]PBR28 is a radioligand of the 18 kD translocator protein, the up-regulation of which is coupled to the level of neuroinflammation in the brain, and [^18^F]flumazenil is a radioligand for GABA_A_-benzodiazepine receptors, whose level mirrors interneuronal activity and eventually cell death, the use of the two radioligands may reveal two critical features of TBI. An up-regulation in the [^11^C]PBR28 uptake triggered by the LFP in the injured (right) hemisphere was noted on day 14, while the uptake of [^18^F]flumazenil was down-regulated on day 14. When comparing the left (contralateral) and right (LFPI) hemispheres, the differences between the two in neuroinflammation were obvious. Our results demonstrate a potential way to measure the molecular alterations in a rodent-based TBI model using PET imaging with [^11^C]PBR28 and [^18^F]flumazenil. These radioligands are promising options that can be eventually used in exploring the complex in vivo pharmacokinetics and delivery mechanisms of nanoparticles in TBI treatment.

## 1. Introduction

Traumatic brain injury (TBI) has recently come under considerable scrutiny owing to extensive human mortality and morbidity associated with war and terrorism [1,2,3]. Several experimental models including fluid percussion [4,5], cortical impact [6,7] and weight drop [8] are widely adapted for rodents to investigate changes at a cellular and molecular level, typically brought about by TBI studied in human beings. Among these TBI models, lateral fluid percussion (LFP) is the most well-endorsed and frequently used design for evaluating mixed focal and diffuse brain injury [9]. This model has the ability to recreate the physiological, pathological as well as behavioral characteristics in rodents under distinct well-controlled clinical conditions. The LFP-induced (LFPI) animal TBI model reproduces lacerations observed in human beings, thus making it a clinically germane model, and enables the surveillance of novel therapeutics that may be adapted to the clinic [5]. Besides, the LFP procedure can be standardized for the employment of variables in TBI and tracking of long-term continual alterations in the brain resulted by primary, as well as secondary injuries [10].

Recent advancements in positron emission tomography (PET) techniques have offered us the prospect of not only qualitatively, but also quantitatively delineating in vivo cellular and molecular transformations triggered by TBI in both humans and animals [11,12]. TBI causes neuroinflammation and a loss of neuronal density [13,14]. In both cases we can identify molecular biomarkers that can be targeted by PET radioligands. To analyse the cellular and molecular alterations of the TBI perceived in humans, an LFPI rodent model was developed. The fluid percussion device creates a laceration through a craniotomy by applying a brief fluid pressure pulse via the intact dura onto the surface of the brain. As the pendulum hits the piston of a fluid reservoir, a pulse is generated. The percussion then leads to a transient dislocation of and damage to the affected nervous tissue in the cerebrum [10].

Over the last few decades, neuroscientists have shown a considerable interest in quantifying neuroinflammation in a living human brain [15]. A common strategy is to use PET radiotracers to target the translocator protein (TSPO) (molecular weight of 18 kDa), a biomarker for neuroinflammation that is up-regulated in activated microglia, as an imaging probe to monitor its uptake and consequently analyse the TSPO expression levels in the brain [16]. Carbon-11 labelled PBR28 is the most commonly used PET radiotracer to quantify TSPO [17] and several studies, using [^11^C]PBR 28, have been published with an aim of visualizing and quantifying neuroinflammation caused by TBI in both human as well as animal models in vivo [18,19,20]. On the other hand, the γ-aminobutyric acid type A (GABA_A_) ionotropic receptor is an ideal biomarker that mirrors neuronal density and integrity alterations [21]. Flumazenil, a selective GABA_A_-benzodiazepine receptor antagonist, when radiolabeled with fluorine-18, is capable of indicating at a high precision the neuronal loss that ensues in various brain disorders and injuries [22]. In this study, [^18^F]flumazenil was applied to examine the GABA_A_-benzodiazepine receptors present in cortical interneurons [23,24].

Immunohistochemistry is a commonly used histopathological technique to correlate the presence of disease-specific biomarkers with the signals obtained from imaging techniques such as PET [25]. The corpus callosum is one of the most frequently affected regions of the cerebrum following the diffuse axonal injury that ensues after TBI [26], and its susceptibility is largely attributed to its localization to the midline of the brain [27]. Recent studies have provided insights into the persistence of neuroinflammatory processes in the corpus callosum due to a chronic increase in reactive microglia in its white matter [28]. The hippocampus is another key brain region that is compromised in the event of TBI and GABA_A_, a receptor of the primary inhibitory neurotransmitter, GABA, exhibits an altered expression of its constituent subunits in post-TBI hippocampi [29].

To date, no systematic study has been carried out using two distinct PET radioligands sequentially on the same day to quantify both neuroinflammation and loss of neuronal density. Here we have examined an LFPI TBI rodent model to study the neuroinflammation and decrease in neuronal density, by injection of PET radioligands that target the respective biomarkers for both phenomena. This investigation focused on whether (i) the two radioligands ([^11^C]PBR28 and [^18^F]flumazenil) could be used in conjunction to dependably measure the in vivo molecular-cellular alterations in a rodent TBI model, and (ii) there are any conceivable variations linked to primary as well as secondary injuries that could be noticed with the two radiotracers—[^11^C]PBR28 and [^18^F]flumazenil. We also aimed to assess whether there are any apparent modifications in both the injured (affected, ipsilateral) and contralateral hemispheres. In addition to the PET imaging that was performed, in vitro immunohistochemical analyses on the corpus callosum and hippocampal sections of the cerebrum were done to validate the results obtained in the PET imaging. To visualize the neuroinflammation in these rat coronal sections, OX-42, a monoclonal antibody indicating the presence of activated microglia [30], was used in the immunohistochemical staining. In this case, postmortem histopathological validation served to test the accuracy and reliability of the quantifying neuroinflammation with [^11^C]PBR 28. According to our knowledge, this is the first investigation examining both neuroinflammation and loss of neuronal density for an LFPI TBI rat model.

Upon the validation of the PET radiotracers through this longitudinal study, they can be potentially employed in future investigations of TBI treatment options using nanomedicine. Nanomaterials present special physicochemical characteristics that promise their unparalleled application in the treatment of various neurodegenerative diseases and neurological injuries [31,32]. In particular, nanotheranostics form a branch of nanomedicine, which integrates both the diagnostic and therapeutic capabilities of nanoparticles into a single agent [33]. The combination of PET imaging capabilities together with the theranostic potential of nanoparticles would not only allow for a better comprehension of their bioavailability, pharmacokinetics, receptor occupancy, and treatment monitoring [34,35], but would also enable the real-time monitoring of the administered nanotheranostics. In addition, nanotheranostic agents may even be radiolabeled to track treatment prognosis [36,37].

An example of nanomedical application that has been studied recently in the therapy of inflammatory brain pathologies, is drug delivery via lipid nanoparticle (LNP) encapsulation to an appropriate target molecule in the inflamed brain. The LNP delivered a therapeutic mRNA to the atypical cerebrovascular endothelium with high precision, ameliorating cerebrovascular edema, a common occurrence in TBI. The LNPs were labelled with indium-111, and their uptake in the cerebrum was monitored using computed tomography (CT) and single-photon emission computed tomography (SPECT) [38]. As such, the possibilities of non-invasive imaging utilizing nuclear medicine imaging methods are widespread and coupling them with nanomedicine unravels a new paradigm in disease treatment. Before PET radiotracers can be combined with nanoparticulate formulations effectively to target neuroinflammation and neuronal loss resulting from TBI, their efficacy would have to be validated in vivo. This forms the main objective of the current study.

## 2. Results

### 2.1. PET Imaging

According to a standard LFP procedure, a combination of focal and diffuse injury was inflicted on the cerebral cortex and hippocampus of the right hemisphere of rats to create a TBI model for the study. The pendulum was placed at a suitable elevation to conduct a fluid pressure pulse through the saline reservoir, and upon its release, unrestrainedly fell to strike a piston. An epidural injection of a miniscule quantity of saline into the closed cranial cavity then generated a brief disturbance to and injury of the neural tissue.

The left hemisphere, since not subject to any injury, served as an internal control for the study. As shown in Figure 1, compared to day 2 post-operation (post-op), there is an increase in the uptake of [^11^C]PBR28 on day 14 due to the LFP in the right hemisphere (injured), while [^18^F]flumazenil uptake was down-regulated on day 14, compared to day 2. The time activity curves (TACs) of the whole brain (Figure 2) also clearly demonstrate that there was a higher [^11^C]PBR28 and lower [^18^F]flumazenil uptake in day 14 as compared to day 2.

Hence, it is clear that the neuroinflammation and the neuron density can be observed in the TBI rodent model using [^11^C]PBR28 and [^18^F]flumazenil PET imaging. When juxtaposing the right and left hemispheres using an area-under-the-curve (AUC) measure, the discrepancies between the two hemispheres for the [^11^C]PBR28 radiotracer were apparent (Figure 3A), whereby the left hemispheres repeatedly displayed lower rates of tracer uptake as compared to the right hemispheres on both days post-op. This is an indication that local increases in neuroinflammation due to the physical impact can be observed in the LFPI TBI rodent model. On the other hand, while [^18^F]flumazenil uptake is slightly higher in the left hemisphere (Figure 3B), the lack of marked changes between the two hemispheres may reflect either the lack of neuron density alterations or the inappropriateness of the radioligand in indicating neuron density changes. This is the first LFPI TBI rat study to evaluate neuroinflammation and loss of neuronal density using two radioligands subsequently on the same day.

From Figure 4A,B, it is evident that the while in all the rats injected with [^11^C]PBR28, the right hemisphere displayed elevated standardized uptake values (SUVs) than the right, 50% of the rats injected with [^18^F]flumazenil demonstrated an opposite pattern. The joint analysis of the SUVs and AUCs presented an apparent up-regulation in [^11^C]PBR28 targeting, and hence an implication of increased neuroinflammation owing to TBI. However, in general, the uptake of [^18^F]flumazenil has seen a down-regulation in the right hemisphere as compared to the left, indicating a possible reduction in GABAergic neuronal density following TBI. Nevertheless, since this pattern is not observable across the cohort of rats used for the study, the down-regulation is not marked and hence not conclusive.

### 2.2. Immunohistochemical Analyses

The immunohistochemistry data unequivocally support the PET imaging results. From Figure 5 we could clearly distinguish the TBI brain from the control brain. The neuroinflammation marker, OX-42, shows a higher binding in the injured hemisphere as compared to the contralateral hemispheres.

We also found that microglial cells become activated after TBI, and the morphology of microglial cells in the post TBI brains was amoeboid in appearance (data not shown). This proves that the higher neuroinflammation appearance is due to LFPI at injured hemispheres.

## 3. Materials and Methods

Twelve Sprague Dawley (SD) rats (male) from InVivos Pte Ltd. (Singapore) were used for this study. The rats were 8–12 weeks old and their weights were in the range of 200–300 g. Approval for the present study (development and validation of preclinical animal models to examine therapeutic response using molecular imaging (PET/MR and SPECT)) was obtained from the SEMC (SingHealth Experimental Medicine Centre, Singapore) Institutional Animal Care and Use Committee (IACUC) (SEMC-981 (DSO_NTU)).

Anhydrous N,N-dimethylformamide (DMF) and anhydrous dimethylsulfoxide (DMSO) were purchased from Fluka (Dorset, UK). The flumazenil and PBR28 precursors were obtained from Pharmasynth (Tartu Estonia, Eesti, Vabariik). Other solvents and chemicals of analytical grade were purchased from Sigma-Aldrich (Singapore). [^18^F]fluoride (no carrier added) was generated in cyclotron (GEMS PETtrace at Singapore Radiopharmaceuticals, RadLink) through a ^18^O(p,n)^18^F reaction by irradiating 16.4 MeV proton to ^18^O-enriched water in a high-pressure small volume target system. Radiosynthesis was remotely conducted in either GE TRACERLAB FXFN (^18^F-synthetic module) or TRACERLAB FXC pro (^11^C-synthetic module).

### 3.1. Preparation for Lateral Fluid Percussion

Isoflurane (2–5%) in oxygen with a flow of 1 Liter/minute was used for inhalational anesthesia in the animals before the surgery was performed. A custom-designed device was used to create the TBI model that comprises a plexiglas cylindrical reservoir filled with distilled water or saline [4,39]. The reservoir has two ends, out of which one contains a plexiglas piston affixed on O-rings, and the other end has a transducer housing with a male Luer-lock opening (inside diameter of 2.6 mm). In this particular design, a pendulum was maintained at a right angle to the ground, abutting the piston at resting position. Sterile saline was used to fill the head cannula to remove any bubbles. Saline thoroughly filled the tubing from the percussion device that was joined to the head cannula by twisting. The pendulum was positioned at an elevation that would cause severe laceration (~50–70 psi) by using the fluid pressure force pulse transmission through the reservoir filled with saline, released, and permitted to fall without restriction to collide with the piston. As a result of the LFP, a mixed focal and diffuse damage was induced in the ipsilateral cerebral cortex as well as in the hippocampus. By virtue of a quick epidural administration of saline in small volume into the closed cranial cavity, a momentary dislocation and contortion of nervous tissue ensued. The head cannula was subsequently expelled from the rat, the burr hole was covered with bone wax lined underneath with hemostatic sponge, and the incision was closed with a suture.

### 3.2. Post-FPI Monitoring Phase

Animals were allowed to recover from the anesthesia. The physiological condition of the animals was monitored for any signs of distress, pain or suffering. Animals were individually housed with food and water available ad libitum after FPI. When and if needed, local antibiotic (Bacitracin) dose was administered. If symptoms of pain or suffering manifested and persisted, no analgesic was given as it could have central nervous system effects and affect cognitive outcome [40]. In this case, the animals were euthanized.

### 3.3. Euthanasia of the Animals

At allocated junctures after FPI, doses of 75 mg/kg and 10 mg/kg of Ketamine and Xylazine respectively, were administered intraperitoneally to the animals to euthanize them. Then the animals were given cardiac puncture to retrieve blood for the analysis of systemic markers; the brains were perfused with fixative (10% neutral buffered formalin), following which they were extracted for further processing and immunohistochemical analyses. Six rats were sacrificed at day 3 and the remaining six rats were sacrificed at day 14 post-FPI.

### 3.4. Perfusion

In order to fully remove blood from the liver and lungs, the animals were subjected to perfusion using Ringer’s solution. Thereafter, fixation was performed using an aldehyde solution, which was made up of 0.075 M lysine, 0.01 M sodium periodate, and 2% paraformaldehyde, which had a pH of 7.4. The rat brains were extracted and preserved in an analogous fixative for two hours following perfusion. Then the brains were transferred to a phosphate buffer solution (0.1 M) comprising sucrose (20%) and kept at 4 °C overnight.

### 3.5. Immunohistochemistry

Frozen coronal brain sections were sliced and washed in PBS buffer. The blocking of the endogenous peroxidase enzyme was carried out by incubating the brain slices for 30 min in PBS-triton solution (PBS-T) comprising 0.1% hydrogen peroxide. Then the slices were washed thrice with PBS-T for 5 min. The immunohistochemistry was then performed by incubating the brain slices for around 18–20 h at room temperature with the OX-42 primary antibody (Sera-Lab MAS370b), diluted at 1:100 with PBS. In the case of control, some slices were incubated without any primary antibodies. Thereafter, the antibodies were detected with the aid of Vectastain ABC kit anti-mouse IgG (PK 4002, Vector Lab) using the peroxide substrate-3, 3′-diaminobenzidine tetrachloride (DAB).

### 3.6. Synthesis of [^11^C]PBR28

[^11^C]PBR28 was synthesized according to a previously reported procedure [41]. [^11^C]CO_2_ produced in Cyclotron was transformed to [^11^C]MeI using a GE TRACERLAB FXC pro radiosynthesis module. Then the [^11^C]MeI was moved through a silver triflate column at 180 °C to produce [^11^C]methyltriflate. The [^11^C]methyltriflate was then gone through 350 mL anhydrous acetone solution containing 0.5 mg of desmethyl-PBR2 and 2 µL 0.5 M tetrabutylammonium hydroxide in methanol as a base. After stirring the reaction mixture for 5 min at room temperature, it was diluted with 1 mL of water and injected to high-performance liquid chromatography (HPLC) (solvent system −10 mmol/L ammonium formate in H_2_O/ACN 40/60 (*v*/*v*), flow rate- 6 mL/minute), Agilent C18 column ( Eclipse XDB, 5 mm, 250_9.4 mm, Santa Clara, CA, USA)). The product peak was collected into a round bottom flask and diluted with 50 mL of water. This solution was then gone through a solid-phase C18 cartridge (Classic SepPak, Waters, Milford, MA, USA) in order to trap the product. After that, the C18 cartridge was washed with 10 mL water and the product ([^11^C]PBR28) was eluted with 1 mL ethanol into a vessel comprising 9 mL saline. This solution was then passed through a millex GV filter (0.22 µm, 33 mm, PVDF, sterile Pall, MA, USA) and finally collected into a 20 mL sterile vial (Adelphi, UK). The quality control analysis of [^11^C]PBR28 was performed using an Agilent radio-HPLC. The radiochemical purity of the compound was 97 ± 2% (n = 7) and radioactive yield was found to be approximately 2 to 4% (clear solution). The specific radioactivity was 105 ± 24 GBq/µmol at the end of the synthesis. The radiochemical identity of the compound was verified through co-elution with standard PBR28.

### 3.7. Synthesis of [^18^F]flumazenil

[^18^F]flumazenil was synthesized according to a previously reported procedure [42]. Cyclotron-produced [^18^F]F^−^ was separated from excess [^18^O]H_2_O by passing through a QMA cartridge (SepPak Light, Waters) which was already conditioned with 10 mL potassium carbonate solution (0.5 M) followed by 20 mL (10 mL × 2) water. The [^18^F]F^−^ was retrieved from the QMA cartridge by eluting with K_2.2.2_ solution (2 mL). This solution was made by dissolving K2.2.2 (9.4 mg, 0.025 mmol) and potassium carbonate (1.7 mg, 0.012 mmol) in 2 mL H_2_O/ACN mixture (4/96 *v*/*v*). The H_2_O/ACN solvent mixture was evaporated until dryness at high temperature (130 °C) under nitrogen flow. To this [^18^F]F^−^/K^+^/K_2.2.2_ complex, a solution of nitro-precursor (4–8 mg) dissolved in anhydrous DMSO or DMF (0.5–1 mL) was added and a nucleophilic fluorination reaction was performed by heating the content at 145 °C for 20 min. After cooling down the reaction mixture, it was diluted with water (1 mL) and injected into a reversed-phase C18 preparative HPLC column. The product peak was accumulated and diluted with 50 mL deionized water. Then this was passed through a solid-phase C18 cartridge (Classic SepPak, Waters, Milford, MA, USA) followed by an elution with 1 mL absolute ethanol. The product was collected in a flask containing 9 mL propylene glycol solution. In the end, the final formulation was collected in a 20 mL sterile vial after passing through a sterile filter. The quality control study was performed by an Agilent radio-HPLC system. The radiochemical purity was 98 ± 1% (n = 6) and radioactive yield was found to be 5 to 7% (clear solution). The specific radioactivity was 740 ± 250 GBq/µmol at the end of the synthesis.

### 3.8. PET Imaging and Data Analysis

Following the LFP procedure on the brain’s right hemisphere, day 2 post operation 3D dynamic PET scans were performed using the nanoScan PET/MRI scanner [43] (Mediso, Budapest, Hungary) at the SingHealth Experimental Medicine Centre (SEMC) in Singapore General Hospital. For [^11^C]PBR28, the injected radioactivity was calculated to be approximately 30 ± 4 MBq for a mass of 0.12 ± 0.03 µg that was injected into the rats. For [^18^F]flumazenil, the injected radioactivity, was calculated to be approximately 18 ± 4 MBq for a mass of 0.005 ± 0.002 µg that was injected into the rats (AtomLab™ 500 Dose Calibrator, Biodex, NY, USA). Correspondingly, the volumes injected into the rats were 220 ± 90 µL and 120 ± 50 µL for [^11^C]PBR28 and [^18^F]flumazenil respectively. The animals, succeeding an initial anatomical MRI scan, were first scanned in the PET scanner with [^11^C]PBR28 for 63 min using a dynamic scan protocol. The second PET radioligand [^18^F]flumazenil was injected 80 min later, and the animals were scanned for a duration of 90 min. The detailed time frames for the respective scan protocols were as follows: 8 × 15 s, 4 × 30 s, 2 × 1 min, 2 × 2 min, 4 × 5 min, 3 × 10 min for [^11^C]PBR28; 8 × 15 s, 4 × 30 s, 2 × 1 min, 2 × 2 min, 4 × 5 min, and 6 × 10 min for [^18^F]flumazenil. Tail vein injections of the radioligands were administered to all the animals. Each rat was injected with [^11^C]PBR28 over a duration of 20 min, followed by administration of [^18^F]flumazenil over 2.5 h (approximately 7 half-lives). The PET datasets were analyzed utilizing the image fusion tool in PMOD (version 3.5; PMOD Technologies, Switzerland), and the Px Rat (W. Schiffer) template was used for regional based analysis [44].

## 4. Future Direction and Conclusions

The quantitative analysis of the image data shows that alterations in TSPO levels, pointing out to neuroinflammatory activities, can be visualized in the TBI rodent model with the aid of [^11^C]PBR28, whereas differences in [^18^F]flumazenil uptake were not prominent. The immunohistochemistry analysis clearly distinguished the control brain tissue from the injured brain tissue, which evidently is in accordance with our PET imaging results. In addition, the PET tracer uptakes correlate with the activation in microglia as demonstrated by immunohistochemistry, implying that [^11^C]PBR28 could potentially be used to longitudinally quantify neuroinflammatory processes following TBI. Despite the fact that our datasets are not yet exhaustive, our results vividly demonstrate that there exists a possibility to quantify the regional changes of neuroinflammation with [^11^C]PBR28 PET imaging, effected by a local physical force on the brain.

However, the display of neuron density changes with [^18^F]flumazenil and PET imaging still requires additional investigations. Since [^18^F]flumazenil is selective to GABAergic neurons, the data obtained would be useful when conducting investigations specific to this subset of neurons. A set of proteins that are more ubiquitously expressed in neural cells include the family of synaptic vesicle proteins 2 (SV2), for which novel radiotracers such as [^18^F]UCB-J, [^18^F]UCB-H and their associated fluorine-18-labelled derivatives are specific [45,46]. A quantification of synaptic density might open up avenues to correlate neuronal density changes. These radiotracers could be worth exploring in future work involving parallel investigations of changes in neuroinflammation and neuronal density in TBI models. Nevertheless, this preliminary study can conceivably pave the way for the conjunction of the aforementioned PET radiotracers with future nanomedical therapies for TBI.

## Figures and Tables

**Figure 1 ijms-22-00951-f001:**
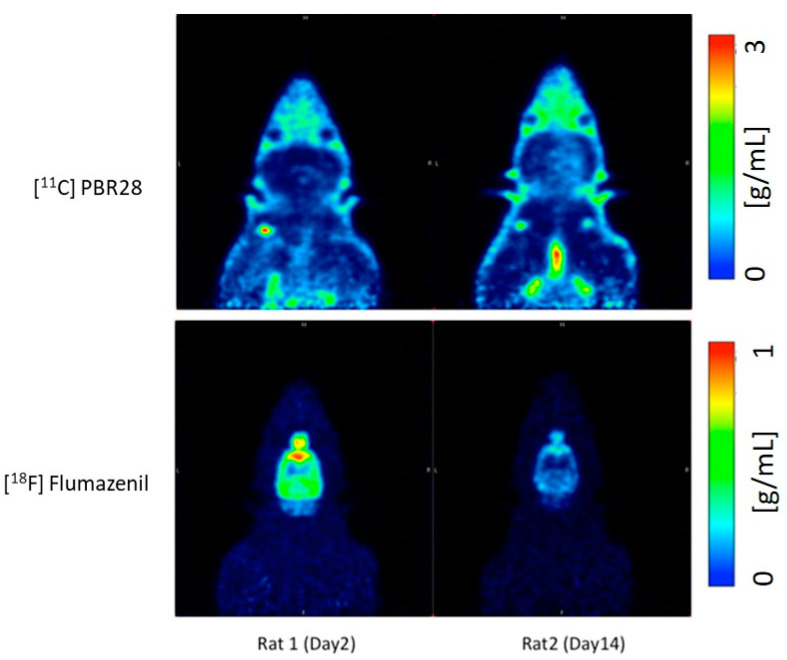
Illustrative PET images of Rat 1 and Rat 2 using [^11^C]PBR28 and [^18^F]flumazenil on day 2 and day 14 post injury.

**Figure 2 ijms-22-00951-f002:**
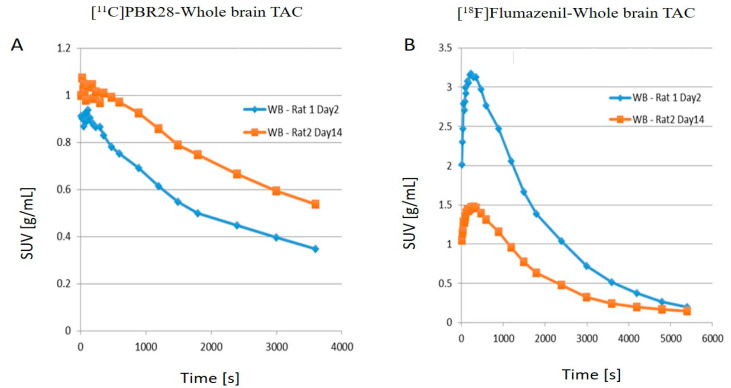
(**A**) [^11^C]PBR28 and (**B**) [^18^F]flumazenil whole brain time activity curves (TACs) for Rat1 day 2 post-op and Rat2 day 14 post-op.

**Figure 3 ijms-22-00951-f003:**
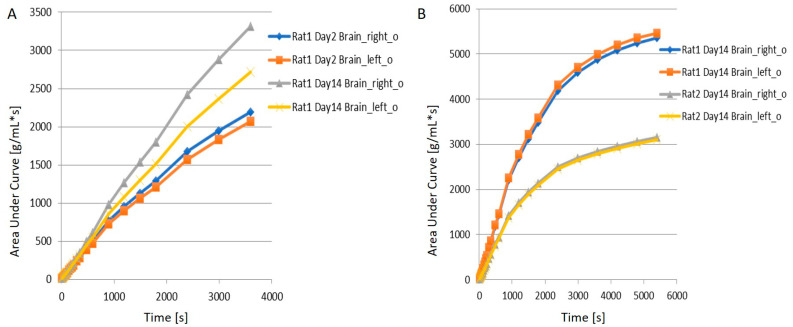
Comparison of tracer uptake for (**A**) [^11^C]PBR28 & (**B**) [^18^F]flumazenil on the left and right hemispheres for Rat1 and Rat2 using area-under-curve (AUC) measures.

**Figure 4 ijms-22-00951-f004:**
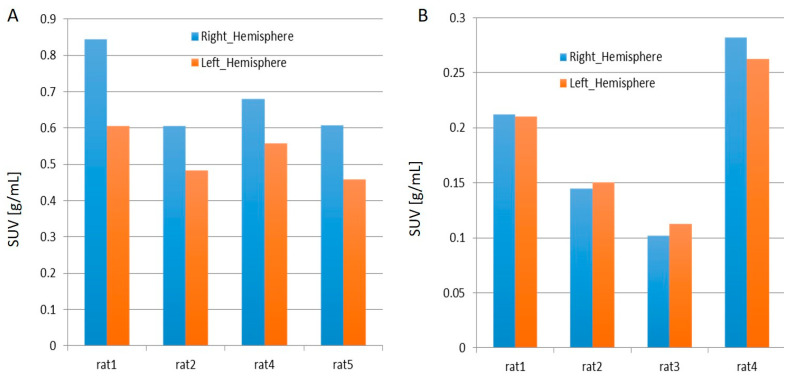
Average standardized uptake values (SUVs) for the last 30 min frames with (**A**) [^11^C]PBR28 and (**B**) [^18^F]flumazenil.

**Figure 5 ijms-22-00951-f005:**
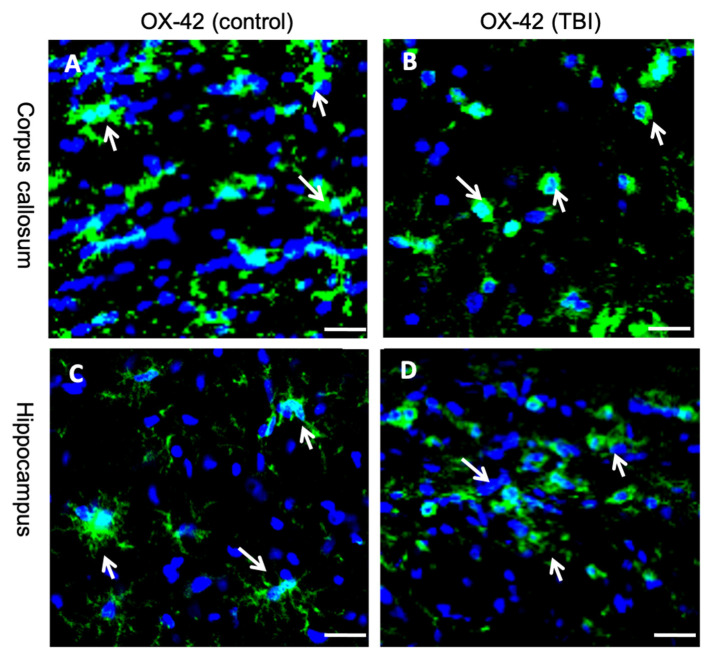
Immunohistochemistry data for corpus callosum ((**A**)—control, (**B**)—TBI) and Hippocampus ((**C**)—control, (**D**)—TBI); Scale bar: 20 µm.

## Data Availability

Data is contained within the article.

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
