# Peer review of "An In Vivo Study of a Rat Fluid-Percussion-Induced Traumatic Brain Injury Model with [11C]PBR28 and [18F]flumazenil PET Imaging"

_ijms, 2021, doi:10.3390/ijms22020951_

Round 1

Reviewer 1 Report

Figures 2 and 3 are duplicates. Based on the caption, figure 3 should be TACs of different hemispheres. The reviewer is surprised to see this low-level mistake from such a reputable group.

Figure 4 contains data from 5 rats. But it is not clear which ones were imaged at 2 days post-op, which ones at 14 days post-op. it seems the neuroinflammation progresses over time and it will be interesting to see if PBR28 can be used to track the changes.

The authors did not observe decreased flumazenil uptake in the ipsilateral hemisphere. The reviewer is curious about why regional uptakes were not analyzed, as the neuronal loss might be restricted in the focal injury site only. It is worth discussing about the limitations of flumazenil targeting only GABAergic neurons. There are new PET imaging probes targeting more ubiquitously expressed synaptic proteins, which could be more sensitive in detecting synapse loss due to the TBI.

The overall study design is under powered to get any definitive conclusion as at which time point post-op, the tracer uptake (either for PBR28 or flumazenil) is different between injury site and contralateral site, or between the injury group and a sham control group (not done in this study).

As the SUV values could vary greatly depending on the actually injected activity or injection route (sometimes, the intravenous injection could be accompanied by partial subcutaneous injection), it is worth trying to normalize the imaging data with a pseudo-reference region, which is not influenced by the injury.

Author Response

General changes/comments:

We have addressed the comments and feedback given by the reviewers. In addition, we have added in three additional references (numbers 32, 45 and 46).

Response to reviewer 1:

  • Figures 2 and 3 are duplicates. Based on the caption, figure 3 should be TACs of different hemispheres.

Response: We have changed figure 3 and its caption accordingly (Lines 173 to 175).

  • Figure 4 contains data from 5 rats. But it is not clear which ones were imaged at 2 days post-op, which ones at 14 days post-op. it seems the neuroinflammation progresses over time and it will be interesting to see if PBR28 can be used to track the changes.

Response: Apologies for the confusion. In the case of rat 3, the injection of PBR28 was not successful, and in the case of rat 5, the flumazenil injection failed. Hence, we have used four rats to be compared in figure 4.

  • The authors did not observe decreased flumazenil uptake in the ipsilateral hemisphere. The reviewer is curious about why regional uptakes were not analyzed, as the neuronal loss might be restricted in the focal injury site only. It is worth discussing about the limitations of flumazenil targeting only GABAergic neurons. There are new PET imaging probes targeting more ubiquitously expressed synaptic proteins, which could be more sensitive in detecting synapse loss due to the TBI.

Response: Thank you for calling attention to this vital point of discussion. We have accordingly included some important additional points about the limitations and flumazenil and the scope for future work, in the final section of the manuscript (Lines 352-360).

  • The overall study design is under powered to get any definitive conclusion as at which time point post-op, the tracer uptake (either for PBR28 or flumazenil) is different between injury site and contralateral site, or between the injury group and a sham control group (not done in this study).

Response: This current model is more authentic because of the measurement on the same animal on another brain hemisphere. Thus, we did not keep a separate Sham control. Moreover, this is only proof of concept or pre-pilot study to test the hypothesis. The different tracer uptakes have been explained in figure 3 (Lines 163 to 169).

  • As the SUV values could vary greatly depending on the actually injected activity or injection route (sometimes, the intravenous injection could be accompanied by partial subcutaneous injection), it is worth trying to normalize the imaging data with a pseudo-reference region, which is not influenced by the injury.

Response: Thanks for a great suggestion from the reviewer. However, in this case a similar injected radioactivity and injection route were applied. In addition, the whole brain time activity curves (TACs) obtained in Fig. 2 were clearly not overlapping so we did not find any reason to normalize the imaging data with a pseudo-reference region.

Reviewer 2 Report

Review of “An in vivo study of a rat fluid-percussion-induced traumatic brain injury model with [11C]PRB28 and [18F]Flumazenil PET imaging

Synopsis

PET studies were performed on rats using a model of traumatic brain injury (TBI). Two radioligands, [11C]PBR28 and [18F]flumazenil, were administered and monitored over the course of 14 days to image and potentially quantify molecular alterations based on TBI. To confirm neuroinflammation, immunohistochemistry technique was performed on the corpus callosum of the brain. The authors conclude that [11C]PBR28 is able to observe neuroinflammation, from day 2 to day 14, whereas [18F]flumazenil did not increase in uptake over the same time frame.

Major Corrections

  • In experimental section, please list the average mass of PBR28 and flumazenil, along with molar activity (Am). Please provide any other analytical parameters obtained from radiosynthesis (i,e, pH, excipients, radionuclidic purity, etc.). Please provide all numbers with as an average with a standard deviation value along with the number of times the run was performed (e.g. 98 ± 5% n = 6). 
  • It is unclear to the reviewer if the same rat was used for each study on different days, or if a new rat (Rat1 v Rat2) was used for each timeline. It is also unclear to the reviewer if the same rats were injected with [11C]PBR28, followed by its decay and then administered [18F]flumazenil. Please state clearly as to how the experiments were conducted for micro-PET and subsequent curved/data generated in this manuscript; please include how many rats (n = 12?) were used for each type of experiment/data generated. Also, please provide the dosed amount of activity injected of each radiotracer (in MBq or equiv. - SI Units) and how the radiotracers were administered. 
  • If possible, an additional experiment which would be useful would be to include a baseline study using normal rats from both tracers to compare/contrast images of the brain when no trauma had occurred. 

Minor issues/Comments

  • Please use Lower case for words such as ‘flumazenil’, ‘positron emission tomography’, ‘day’, ‘flourine-18’ 
  • Line 35 – The phrase “in exploring the complex in vivo complex pharmacokinetics and delivery mechanisms…” is confusing, please consider rewriting/rewording  
  • Line 37 to 39 – Please lowercase all keywords used, except for all acronyms  
  • Introduction (Section 1) is very well written and informative 
  • Please check use of words and phrases (e.g. ‘Till date’ should read ‘To date’) 
  • Section 3 – It is stated twelve (12) SD rats were used for this study; however, in the discussion section, only two of the rat's data was evaluated. Is this an average of several rats? If so what was the standard deviation from the other studies? 
  • Lines 269 and 290 – Please do not use word ‘around’. Please provide analytical data to support purity and quality control of the formed radiotracers. See comment above in regard to analytical data to ben provided for both radiotracers. 
  • Please refer/use the nomenclature as stated in the guideline https://doi.org/10.1016/j.nucmedbio.2017.09.004. (H.H. Coenen, et. al.; Nuclear Medicine and Biology, 2017, 55, v-xi.). As an example, please reframe from using ‘[18F]F-‘ and ‘111In’ in this format; please follow this for all nomenclature discussed in aforementioned reference in your manuscript. 
  • Please provide more discussion as to how the experiments were conducted in the Results Section. For instance, please provide the amount of radiotracer administered (MBq) for each rodent, how these tracers were delivered, (i.e. tail vein injection), what timepoint scanning on the micro-PET takes place, etc. 
  • Please discuss in the Section 2, Results, briefly describe the LFP procedure. 
  • Please elaborate on Figure 4. The reviewer may be missing how this data is compared with the values obtained from Figure 3? 
  • Please explain the difference in Figures 2 & 3. Is this an identical experiment? 
  • A suggestion for Figures 2 and 3 would be to convert the x-axis to minutes instead of seconds as the discussion in the paper is referenced to 30-min time points. 
  • In Section 3. Materials, please provide the dose calibrator make and model used to obtain amount of radioactivity determined for each study. Please provide the gamma counter used in the TAC curves.  
  • Line 229 – Please capitalize ‘Six’ 
  • In Section 3.8 – Please provide the amount of radioligand injected (in MBq) into each subject. Please provide the total volume of the solution along with the mass of carbon-12 and fluorine-19 ligand injected determined from HPLC analysis.  
  • Please use superscript abbreviations for carbon-11, (11C), fluorine-18 (18F) and oxygen-18 (18O). Please use lower case in the abbreviation section. Please follow H.H. Coenen, et. al.; Nuclear Medicine and Biology, 2017, 55, v-xi when referring to isotopes. 
  • Reference #3 – Please change title of journal to ‘New England Journal of Medicine’. Please check spelling and capitalization for all references, as this error appears in several other references (Reference # 10, 11, 13, 14, 15, 16, 24, 27, 29, 30, 32, 33, 34, 35, 40, 41, and 43). 
  • Please check Reference #12, unable to determine what Journal Name is associated with reference. 
  • Reference #14, please add end of page number, ‘142’. 
  • Reference #25, please add end of page number, ‘345’. 
  • Reference #26, please add end of page number, ‘E10’. 
  • Reference #30, please adjust to read as follows: ‘Journal of Anatomy 1992, 180, (2), 333-342.’ 
  • Reference # 38, please adjust to read as follows: ‘Current protocols in neuroscience2004, Chapter 9, Unit 9.2, 11.’ 

Author Response

Major Corrections

  • In experimental section, please list the average mass of PBR28 and flumazenil, along with molar activity (Am). Please provide any other analytical parameters obtained from radiosynthesis (i,e, pH, excipients, radionuclidic purity, etc.). Please provide all numbers with as an average with a standard deviation value along with the number of times the run was performed (e.g. 98 ± 5% n = 6). 

Response: We have added in the necessary details. Please refer to lines 291-292, lines 314-316 and lines 321-324.

  • It is unclear to the reviewer if the same rat was used for each study on different days, or if a new rat (Rat1 v Rat2) was used for each timeline. It is also unclear to the reviewer if the same rats were injected with [11C]PBR28, followed by its decay and then administered [18F]flumazenil. Please state clearly as to how the experiments were conducted for micro-PET and subsequent curved/data generated in this manuscript; please include how many rats (n = 12?) were used for each type of experiment/data generated. Also, please provide the dosed amount of activity injected of each radiotracer (in MBq or equiv. - SI Units) and how the radiotracers were administered. 

Response: Thank you so much for the reviewer’s valuable question! Since the half-life of the ligands are very different (PBR28 vs Flumazenil); the same rat was injected with carbon-11 (20 minutes) followed by fluorine-18 Flumazenil (we injected after 7 half-lives of PBR28 over a duration of 2.5hrs). We have included this in lines 334 to 335. Even though we created FPI TBI model on several rats at one time, we had only used few animals for the imaging study and the rest of them for  Immunohistochemistry (IHC) experiments in order to keep the time point sampling. We have included the dosed amount of injected radioactivity in MBq (lines 321-324) and mass injected and also included the mode of administration (tail-vein injection) in lines 333-334.

  • If possible, an additional experiment which would be useful would be to include a baseline study using normal rats from both tracers to compare/contrast images of the brain when no trauma had occurred. 

Response: Thank you very much for this question. Unfortunately, we were not able to conduct an additional experiment in this short time span. However, the FPI TBI model is very well studied by several researchers in rats (A Lateral Fluid Percussion Injury Model for Studying Traumatic Brain Injury in Rats; Methods Mol Biol. 2018;1717:27-36. doi: 10.1007/978-1-4939-7526-6_3.). Please refer to lines 142-143 on the note that the left hemisphere served as an internal control as it was not subject to any injury.

Minor issues/Comments

  • Please use Lower case for words such as ‘flumazenil’, ‘positron emission tomography’, ‘day’, ‘flourine-18’ 

Response: Thank you for this note and we have made the necessary changes.

  • Line 35 – The phrase “in exploring the complex in vivo complex pharmacokinetics and delivery mechanisms…” is confusing, please consider rewriting/rewording  

Response: Thank you for this comment and we have rewritten the phrase to be clearer, and the text is highlighted (lines 34-35).

  • Line 37 to 39 – Please lowercase all keywords used, except for all acronyms  

Response: We have made the necessary changes.

  • Introduction (Section 1) is very well written and informative 

Thank you for the comment!

  • Please check use of words and phrases (e.g. ‘Till date’ should read ‘To date’) 

Response: We have checked these and made changes, where needed (line 90).

  • Section 3 – It is stated twelve (12) SD rats were used for this study; however, in the discussion section, only two of the rat's data was evaluated. Is this an average of several rats? If so what was the standard deviation from the other studies? 

Response: Even though we created the FPI TBI model on several rats at one time, we have used only a few animals for the imaging study and the rest of them for the immunohistochemical (IHC) study in order to keep the time point sampling.

  • Lines 269 and 290 – Please do not use word ‘around’. Please provide analytical data to support purity and quality control of the formed radiotracers. See comment above in regard to analytical data to be provided for both radiotracers. 

Response: Thank you for pointing this out. We have corrected this for both PBR28 and flumazenil (lines 291 and 314).

  • Please refer/use the nomenclature as stated in the guideline https://doi.org/10.1016/j.nucmedbio.2017.09.004. (H.H. Coenen, et. al.; Nuclear Medicine and Biology201755, v-xi.). As an example, please reframe from using ‘[18F]F-‘ and ‘111In’ in this format; please follow this for all nomenclature discussed in aforementioned reference in your manuscript. 

Response: Thank you for this crucial comment on nomenclature. We have made the necessary changes as per the guideline. We have checked the guideline and the way to label a radiolabelled ion is as follows: <Likewise, ‘76Br-’ (the bromine-76 anion) is more accurately described in texts by the terms [76Br]bromide ion or [76Br]Br-, and by analogy, [177Lu]Lu3+ is correct for the description of the [177Lu]lutetium cation rather than ‘177Lu3+’> - (taken from H.H. Coenen, et. al.; Nuclear Medicine and Biology201755, v-xi.).

Therefore, we have retained ‘[18F]F-‘  in our manuscript

  • Please provide more discussion as to how the experiments were conducted in the Results Section. For instance, please provide the amount of radiotracer administered (MBq) for each rodent, how these tracers were delivered, (i.e. tail vein injection), what timepoint scanning on the micro-PET takes place, etc. 

Response: Thank you for this comment. We have mentioned the amounts of radiotracers administered (lines 321-324), as well as the mode of delivery and timepoint scanning on the micro-PET (lines 330 to 335).

  • Please discuss in the Section 2, Results, briefly describe the LFP procedure.

Response: We have briefly described the LFP procedure at the start of the results section (please refer to lines 135-140).

  • Please elaborate on Figure 4. The reviewer may be missing how this data is compared with the values obtained from Figure 3? 

Response: We have added in an explanation of Figure 4 (please refer to lines 180-189).

  • Please explain the difference in Figures 2 & 3. Is this an identical experiment? 

Response: Thank you for pointing this out. We have changed figure 3 and its caption accordingly (lines 173 to 175).

  • A suggestion for Figures 2 and 3 would be to convert the x-axis to minutes instead of seconds as the discussion in the paper is referenced to 30-min time points. 

Response: Thank you for the note. We have added in a detailed scan protocol in the methods section (lines 330 to 333).

  • In Section 3. Materials, please provide the dose calibrator make and model used to obtain amount of radioactivity determined for each study. Please provide the gamma counter used in the TAC curves.  

Response: We have accordingly added in the dose calibrator make and model in lines 324-325. We have not provided a gamma counter data as the software we have used in the analysis of PET data, PMOD, is embedded with the activity measurement capability to calculate the TACs values (lines 336).

  • Line 229 – Please capitalize ‘Six’ 

Response: We have made the change (line 251).

  • In Section 3.8 – Please provide the amount of radioligand injected (in MBq) into each subject. Please provide the total volume of the solution along with the mass of carbon-12 and fluorine-19 ligand injected determined from HPLC analysis.  

Response: We have provided the amount of radioligand injected, the total volume and masses of the radioligand determined in lines 321-326.

  • Please use superscript abbreviations for carbon-11, (11C), fluorine-18 (18F) and oxygen-18 (18O). Please use lower case in the abbreviation section. Please follow H.H. Coenen, et. al.; Nuclear Medicine and Biology201755, v-xi when referring to isotopes. 

We have adhered to the guideline and made the necessary changes.

  • Reference #3 – Please change title of journal to ‘New England Journal of Medicine’. Please check spelling and capitalization for all references, as this error appears in several other references (Reference # 10, 11, 13, 14, 15, 16, 24, 27, 29, 30, 32, 33, 34, 35, 40, 41, and 43). 

Response: We have made the necessary changes.

  • Please check Reference #12, unable to determine what Journal Name is associated with reference. 

Response: We have added the name of the journal to this reference.

  • Reference #14, please add end of page number, ‘142’. 

Response: We have added the end of page number.

  • Reference #25, please add end of page number, ‘345’. 

Response: We have added the end of page number.

  • Reference #26, please add end of page number, ‘E10’. 

Response: We have added the end of page number.

  • Reference #30, please adjust to read as follows: ‘Journal of Anatomy 1992, 180, (2), 333-342.’ 

Response: We have adjusted accordingly.

  • Reference # 38, please adjust to read as follows: ‘Current protocols in neuroscience2004, Chapter 9, Unit 9.2, 11.’ 

Response: We have adjusted accordingly.

Round 2

Reviewer 2 Report

To the Authors,

Thank you for making adjustments to the original manuscript.

The Reviewer has no further comments/suggestions.